# Genome-Wide Characterization and Evolutionary Expansion of Poplar NAC Transcription Factors and Their Tissue-Specific Expression Profiles under Drought

**DOI:** 10.3390/ijms24010253

**Published:** 2022-12-23

**Authors:** Lu Meng, Siyuan Chen, Dawei Li, Minren Huang, Sheng Zhu

**Affiliations:** 1College of Biology and the Environment, Nanjing Forestry University, Nanjing 210037, China; 2Key Laboratory of Forest Genetics and Biotechnology, Ministry of Education of China, Co-Innovation Center for the Sustainable Forestry in Southern China, Nanjing Forestry University, Nanjing 210037, China

**Keywords:** *Populus*, *NAC* genes, drought-responsive, gene duplication events

## Abstract

The NAC (NAM, ATAF1/2 and CUC2) is a large gene family of plant-specific transcription factors that play a pivotal role in various physiological processes and abiotic stresses. Due to the lack of genome-wide characterization, intraspecific and interspecific synteny, and drought-responsive expression pattern of *NAC* genes in poplar, the functional characterization of drought-related *NAC* genes have been scarcely reported in *Populus* species. Here, we identified a total of 170 *NAC* domain-containing genes in the *P. trichocarpa* genome, 169 of which were unevenly distributed on its nineteen chromosomes. These *NAC* genes were phylogenetically divided into twenty subgroups, some of which exhibited a similar pattern of exon–intron architecture. The synteny and Ka/Ks analysis indicated that the expansion of *NAC* genes in poplar was mainly due to gene duplication events occurring before and after the divergence of *Populus* and *Salix*. Ten *PdNAC* (*P. deltoids × P. euramericana* cv.’Nanlin895’) genes were randomly selected and cloned. Their drought-responsive expression profiles showed a tissue-specific pattern. The transcription factor PdNAC013 was verified to be localized in the nucleus. Our research results provide genomic information for the expansion of *NAC* genes in the poplar genome, and for further characterizing putative poplar *NAC* genes associated with water-deficit.

## 1. Introduction

NAC (*Petunia* NAM [1], and *Arabidopsis* ATAF1/2 and CUC2 [2] is one of the largest multigene families of plant-specific transcription factors (TFs), which are the important regulatory proteins in gene expression networks [3]. The NAC proteins are roughly partitioned into two parts, including a hypervariable transcription regulatory domain (TRD) at its C-terminus and a highly conservative NAC domain of nearly 150 amino acids at the N-terminus [4]. The N-terminal domain (NTD) of NAC proteins functions as a DNA-binding domain (DBD) which regulates the expression level of target genes by interacting with the *cis*-element at their promoters [5]. In addition to DNA-binding, the NAC domains are associated with nuclear localization and NAC dimer formation [4]. The NAC TFs have been found to regulate the expression of genes involved in the response of plant against diverse abiotic stresses, such as drought, high salinity, high (heat) and low (cold) temperature [6,7].

Identification of NAC TFs at the genome-wide level has been documented in the genomic sequences of angiosperm species, such as 75 in *Oryza sativa* [8], 105 in *Arabidopsis thaliana* [8], 74 in *Vitis vinifera* [9], 189 in *Eucalyptus grandis* [10], and 93 in *Solanum lycopersicum* [11]. Meanwhile, recent studies suggested that plant NAC TFs are related to the transcriptional regulation of drought-induced genes, water movement in plants, and the improvement of drought tolerance. For example, overexpression of the *OsNAC10* gene enhanced drought resistance of transgenic rice plants with an increased grain yield [12]. *ANAC016* in *A. thaliana* has an inhibitory effect on the expression of *AREB1* (ABSCISIC ACID-RESPONSIVE ELEMENT BINDING PROTEIN1) gene, which encodes an important regulator in ABA-dependent signaling pathway [13]. Two *Arabidopsis VND* (vascular related NAC-domain) genes, including *VND6* and *VND7*, are involved in xylem differentiation and cell fiber differentiation [14]. The NAC TFs in *Physcomitrella patens* are related to water transport in cellular tissues [15]. Overexpression of the *CarNAC3* gene from *Cicer arietinum*, can lead to an increase in proline content and antioxidant enzyme activities, and a decrease in MDA (malondialdehyde) concentration of transgenic poplar NL895 (*Populus deltoides × P. euramericana* cv. ′Nanlin895’) [16]. In addition, only a handful of drought-related poplar NAC TFs, such as *PeNAC034*, *PeNAC045, PeNAC036* and *PeNAC122* genes of *P. euphratica*, have been functionally characterized to date [17,18].

The frequency of extreme meteorological events and the change of rainfall pattern aggravate the degree of drought at a regional scale [19]. Severe droughts have caused widespread tree mortality across many forest biomes with profound impact on the function of the ecosystem and the carbon balance [20,21]. The *Populus* species, as a settled and woody perennial species, may have evolved a sophisticated regulatory mechanism by which genes are regulated in response to water deficit occurring over the poplar lifetime. However, the regulating role of NAC TFs in poplar in response to drought stress remains largely unknown. This is, in part, due to the lack of the basic information of poplar NAC TFs, such as their coding sequence, physical and chemical properties, subcellular localization, drought stress-responsive expression profiling, and divergent evolutionary pattern.

In this research, NAC transcription factors in the whole *P. trichocarpa* genome were identified, in which comprehensive bioinformatics analyses were performed, including the physicochemical attribution, chromosomal distribution, phylogenetic classification, and intraspecific and interspecific collinearity detection. Ten poplar NAC genes were randomly selected and cloned from the genome of the poplar clone NL895. Their drought-responsive expression patterns were analyzed in three tissues (e.g., root, stem and leaf) of poplar NL895 using quantitative real-time PCR (qRT-PCR). Finally, the subcellular localization of PdNAC013 protein, which was encoded by one of the selected ten NAC genes, was verified in poplar NL895 protoplasts.

## 2. Results

### 2.1. Identification of PtrNAC Family Genes in Poplar

Here, we identified a total of 289 putative NAC (Pfam No.PF02365) domain-containing transcripts located in the entire *P. trichocarpa* genome. These transcripts were transcribed from 170 gene loci of the *P. trichocarpa* genome. The 170 *NAC* genes were reassigned as *PtrNAC001*-*PtrNAC170* according to their corresponding chromosome number and physical location. Only one (*PtrNAC170*) of all *PtrNAC* genes was not located on any of the nineteen *P. trichocarpa* chromosomes.

The PtrNAC proteins ranged from 122 amino acid (aa) (PtrNAC125) to 1524 aa (PtrNAC170) in size, with an average length of 350.71 aa. Their corresponding molecular weights were between 14.84 kDa (PtrNAC125) and 174.42 kDa (PtrNAC170). Their predicted pI (isoelectric point) varied from 4.28 (PtrNAC138) to 10.73 (PtrNAC021). The coding sequences of these *PtrNAC* genes had an average GC content of 43.89%, with a relatively wide range of from 37.84% (PtrNAC120) to 52.44% (PtrNAC055). Detailed information on these data was summarized in Appendix A.

### 2.2. Chromosomal Distributions and Collinearity Analysis of PtrNACs

Of all 170 *PtrNAC* genes, 169 were located on all nineteen chromosomes of *P. trichocarpa*. The 169 *PtrNAC* genes shared an uneven chromosomal distribution (Figure 1). The total number of *PtrNAC* genes per chromosomes varied from four (Chromosome 18, Chr18) to sixteen (Chr01). The average number of *PtrNACs* per 10 Mb (megabase pair) also differed among nineteen chromosomes in *P. trichocarpa*, ranging from 2.36 (Chr18) to 6.87 (Chr14). It seemed obvious that *PtrNAC* genes were not evenly distributed over a few chromosomes. Three NAC gene-rich clusters were found in Chr02 (*PtrNAC025*-*PtrNAC029*), Chr06 (*PtrNAC060*-*PtrNAC067*) and Chr14 (*PtrNAC129*-*PtrNAC134*). In addition, there were eight tandem duplicated gene pairs distributed on eight chromosomes, including Chr01 (*PtrNAC011* and *PtrNAC012*), Chr02 (*PtrNAC022* and *PtrNAC023*), Chr06 (*PtrNAC063* and *PtrNAC064*), Chr09 (*PtrNAC089* and *PtrNAC090*), Chr12 (*PtrNAC112 PtrNAC113*), Chr14 (*PtrNAC132* and *PtrNAC133*), Chr16 (*PtrNAC146* and *PtrNAC147*) andand Chr19 (*PtrNAC168* and *PtrNAC169*).

Gene duplication events, such as tandem duplication and whole-genome duplication (WGD), were considered as a common process during plant evolutionary process, and could lead to the expansion of multigene families [22]. In order to further investigate the diversity and evolution of NAC gene family members in poplar, we identified the replication events of 170 *PtrNAC* genes during evolutionary innovation. A total of 117 *PtrNAC* genes were identified to be derived from WGD events, suggesting that WGD contributed mainly to the expansion of the NAC gene family in poplar.

The chromosomal synteny analysis discovered a total of 90 collinear gene pairs (Figure 1, Appendix A). The synonymous (Ks) and non-synonymous (Ka) ratio of all duplicated *PtrNAC* gene pairs were less than 1, demonstrating that *PtrNAC* members had experienced purifying selective pressure and their functions might be conserved after the expansion of *NAC* genes in poplar [23,24]. The divergence time of homologous *PtrNAC* pairs could be estimated based on Ks-based distribution. Their divergence times ranged from 9.5 to 208.09 million years ago (Mya). The divergence times for 55 out of all 90 collinear gene pairs were less than 31 Mya. This showed that the expansion of *PtrNACs* occurred after and before the split between *Populus* and *Salix* (45 Mya) [25].

To further elaborate the evolutionary relationship of the NAC gene family, we constructed interspecific collinear relationships between *P. trichocarpa* and four other plants, including *Salix purpurea*, *Eucalyptus grandis*, *Vitis vinifera* and *Arabidopsis thaliana* (Figure 2). A total number of 208, 90, 84 and 70 *PtrNACs* shared a collinear relationship with that of *S. purpurea*, *E.* grandis, *V*. *vinifera* and *A. thaliana*, respectively. The number of orthologous NAC gene pairs between *P. trichocarpa* and *S. purpurea* were more than two-fold that of homologous gene pairs between *P. trichocarpa* and the remaining three species, indicating that a large part of *PtrNAC* genes might be arisen after the divergence of the *Populus* genus from the Salicaceae family.

### 2.3. Phylogenetic Tree and Gene Structures of PtrNACs

To explore the evolutionary relationships of *PtrNACs*, a total of 170 *PtrNAC* genes were phylogenetically categorized into twenty subfamilies in the maximum likelihood (ML) tree. These subfamilies were designated alphabetically as A through T. The number of *PtrNAC* genes in these subfamilies were between two (subfamily D) and twenty-two (subfamily A) (Figure 3). Moreover, the eight tandem duplicated gene pairs belonged to the eight subfamilies distributed on the eight chromosomes A, F, G, K, R, C, K and O, respectively.

To further understand the characteristics of the *PtrNAC* genes, we investigated the exon–intron patterns of *PtrNAC* genes (Figure 4). A total of 74 (44%) and 95 (56%) *PtrNAC* genes were transcribed at the forward (+) and reverse (−) strand of all *P. trchocarpa* chromosomes. Exons or introns were unequally distributed over *PtrNAC* genes, but most related members of the same subfamily had similar intron/exon structures. For example, all *PtrNAC* genes contained introns except for all members belonging to the subfamily G, five genes (e.g., *PtrNAC165, PtrNAC153, PtrNAC135, PtrNAC104* and *PtrNAC073*) belonging to the subfamily H, and *PtrNAC136* in the subfamily R. In contrast, each of the six subfamilies (e.g., N, L, K, J, S and P) shared three exons pattern. The exons in the *PtrNAC* genes had an average number of 3.34. Two *PtrNAC* genes, including *PtrNAC083* and *PtrNAC107*, possessed the maximum number of eight exons.

### 2.4. Cis-Element Analysis of PtrNAC Promoters

Transcription factors have a close association with the regulation of gene expression via binding the promotor region of their target genes [26]. Here we analyzed the *cis*-elements in the upstream region of 289 *PtrNAC* transcripts (Figure 5, Appendix A). *Cis*-acting elements that appeared at the promoter region of more than 150 of all 289 transcripts could roughly fall into three categories: light-responsive elements, phytohormone-responsive elements, stress-responsive elements. The *cis*-acting elements associated with phytohormone could be divided into three subcategories, including ABA-related elements (ABRE (abscisic acid-responsive element) and G-box) [27], ethylene-related elements (ERE (ethylene-responsive element)) [28] and MeJA-related elements (CGTCA-motif and TGACG-motif) [29]. ABA and MeJA were two major phytohormones related to an increase of ROS (reactive oxygen species) and NO (nitric oxide) and a regulator of stomatal closure [30]. The *cis*-elements belonging to stress-responsive element were consisted of STRE (stress response element), TC-rich repeat, MYB and MYB-like sequence. A large proportion of *PtrNACs* contained MYB binding sites in their promoter region, suggesting that MYB might be a major kind of transcription factors binding to *PtrNACs* [31]. This result indicated that *PtrNAC* genes were implicated in abiotic stress response (e.g., drought), and plant growth and development.

### 2.5. Cloning of Poplar NAC Genes Cloning and Analysis of PtrNAC Expression Patterns in Response to Drought Stress

Based on the RNA-seq profiling of *PtrNAC* genes under drought stress, ten *PtrNAC* genes were randomly selected for cloning these genes from the NL895 poplar. They were named according to the homology with their corresponding genes in *P. trichocarpa*. The *NAC* gene sequences of the NL895 poplar were compared with that of *P. trichocarpa* and I-69 poplar (*Populus deltoides* bartr. cv. ‘Lux’). Each of these genes shared a pairwise sequence identity of nearly 99% between the three *Populus* sepcies.

The external environment stimulus had a major influence on plant growth and development, and the plant could respond to environmental stress by regulating crucial molecular processes. The RNA-seq profiling data provided a drought-responsive expression pattern of *PtrNAC* genes in different tissues of *P. trichocarpa*. The majority of NAC genes were induced by drought stress in poplar. Among them, ten genes were selected for cloning genes from the NL895 poplar, and their drought-responsive expression pattern in *P. trichocarpa* had been shown in Figure 6A. The RNA-seq profile analysis showed that the drought-responsive expression pattern of these *PtrNAC* genes could be a marked difference among three tissues, including roots, stems and leaves.

Thus, to investigate whether the ten *PdNAC* genes were implicated in poplar responses to drought, we adopted qRT-PCR to analyze their tissue-specific patterns under drought stress. As shown in Figure 6B, the expression levels of the *PdNAC*s changed after 3% (*w*/*v*) PEG6000 treatment. The results shown that the drought-responsive expression patterns of a few *PdNACs* were the difference among the leaf, root and stem tissues. In the leaf tissues, drought-induced expression of all but *PdNAC027* genes shared a single-pulse pattern. The expression abundance of *PdNAC013*, *PdNAC86* and *PdNAC105* genes reached the peak at 72 h after PEG6000 treatment. The expression levels of *PdNAC013* and *PdNAC105* genes increased by nearly 85-fold and 20-fold at 72 h compared to control group (0 h), respectively. The highest expression level of *PdNAC049*, *PdNAC55* and *PdNAC78* genes occurred at 24 h after PEG6000 treatment. In the root tissues, drought-induced expression of all but *PdNAC055* genes could be clustered into three patterns, including single-peak (*PdNAC013, PdNAC049, PdNAC086, PdNAC095* and *PdNAC105*), double-peak (*PdNAC021, PdNAC027* and *PdNAC028*) and triple-peak (*PdNAC078*) pattern. In the stem tissues, only three genes (*PdNAC013*, *PdNAC055* and *PdNAC105*) exhibited a single-pulse pattern in response to drought stress, and genes (*PdNAC021*, *PdNAC028*, *PdNAC049* and *PdNAC078*) showed a double-peak pattern. In short, transcriptional time series of some *PdNAC* genes under PEG6000 treatment might have an impulse pattern [32].

### 2.6. Subcellular Localization Prediction and Verification

Here, we predicted the subcellular localization of 289 PtrNAC transcripts, more than 240 of which were predicted to be localized in the nucleus (Appendix A). The PtrNAC013 and its orthologous protein PdNAC013 in hybrid poplar NL895 were predicted as nuclear-located proteins. To validate further the subcellular localization of PdNAC013 protein, we conducted the transient expression of a GFP-tagged *PdNAC013* constructed in protoplasts of NL895 poplar, with expressing a red nuclear marker D53-mCherry protein [33]. The pBI221::*PdNAC013* fusion was observed microscopically only to localize in the nucleus with the red nuclear marker D53-mCherry protein (Figure 7). In contrast, pBI221-GFP was ubiquitously distributed without specific localization in the cell. The results showed that PdNAC013 was a protein localized clearly in the nucleus.

## 3. Discussion

In this study, we identified a total of 170 *NAC* genes in *P. trichocarpa* genome, which were more than the number of NAC family genes in *O. sativa* (75) [8], *A. thaliana* (105) [8], *V. vinifera* (74) [9] and *S. lycopersicum* (93) [11]. It was obvious that NAC family genes had expanded in *P. trichocarpa*. The intraspecific and interspecific collinearity analysis suggested that the expansion of NAC genes in poplar was caused mainly by WGD events. Recent studies also suggest that WGD event is a major driving force for expansion of multigene family in plant genome [34,35,36]. The expansion of NAC gene family driven by WGD has also been documented in *Pyrus pyrifolia* [37] and *Zea mays* [38]. In addition, *PtrNACs* showed diverse expression, indicating that modifications, such as point mutations, might occur in regulatory regions of duplicated genes, which affected the function and expression patterns [39,40].

*Populus* and *Salix* are two sister lineages which shared common WGD events (named salicoid *duplication*) happening in approximately 58 Mya [41,42]. This was consistent with our interspecific collinearity results that the number of orthologous NAC gene pairs between *P. trichocarpa* and *S. purpurea* were more than twice those of between *P. trichocarpa* and the remaining three species (*E. grandis*, *V. vinifera* and *A. thaliana*). Likewise, the analysis of Ks-based distribution showed that the divergence times for more than half of homologous *PtrNAC* pairs were less than 31 Mya. These results suggested that the expansion of *PtrNACs* occurred after and before the separation time of *Populus* and *Salix* (45 Mya) [25]. Additionally, it is worth noting whether the young *PtrNACs* arisen after the divergence of *Populus* and *Salix* play a role in the adaptation of *Populus* species to their local environment (e.g., drought, salinity and heat) [43,44].

Based on the RNA-seq profile of *PtrNACs* under drought treatment, ten *NAC* (*PdNAC*) genes were randomly selected and cloned from *P. deltoids × P. euramericana* cv.’NL895’. The promoter region of these *NAC* genes contained many *cis*-acting elements related to drought stress, such as ABRE, DRE and G-box binding sites [45,46,47]. Previous studies have shown that NAC transcription factor binds to *cis*-acting elements in the downstream promoter region of drought-related genes, thereby activating the expression of this gene and improving drought resistance in plants [48,49,50]. For example, TaABFs and TaAREB3 can bind to *cis*-acting ABA-responsive element (ABRE elements) of *TaSNAC8-6A* and *TaNAC48* promoter and activated their expression in wheat, thus enhancing drought tolerance of plants [51,52]. *ONAC066* could bind the JBS-like (JBSL) *cis*-element in the promoter region of *OsDREB2A* gene to positively regulate drought and oxidative stress response [53].

Therefore, we analyzed the drought-responsive expression patterns of ten *PdNAC* genes in three tissues of poplar NL895. Six *PdNAC* genes had sequence similarity to *ATAF1* (*ANAC002*) and *RD26* (*ANAC072*) genes, which were found to be drought-responsive *NAC* genes in *A. thaliana* [54]. *ATAF1* is an *Arabidopsis* NAC transcription factor that regulates ABA synthesis [55,56]. *OsNAC6*, a member of ATAF family, can mediate the adaptation of plant root structure and up-regulate the expression of drought-resistant genes, which enhances the drought tolerance of rice plants [57,58,59]. Four poplar NAC genes, including *PdNAC055* (Potri.005G180200), *PdNAC021* (Potri.002G081000), *PdNAC049* (Potri.005G069500) and *PdNAC078* (Potri.007G099400), shared high sequence similarity to *ATAF1* from *A. thaliana*. It seemed obvious that the expression levels of the four *PdNAC* genes were induced by drought stress. *PagNAC045 (*Potri.007G099400*)*, which was isolated from *P. alba* × *P. glandulosa* cv.84K, might be implicated with NaCl and ABA-mediated stresses [60]. However, whether the four *PdNAC* genes were implicated with ABA remains unknown.

*ANAC072 /RD26* is a positive regulator of ABA and drought stress [61]. The gene expression of *ANAC072 /RD26* can up-regulated under drought and salt stress, and the plants showed obvious drought resistance [61,62]. *PdNAC105|PtrNAC105* (Potri.011G123300) and *PdNAC013|PtrNAC013* (Potri.001G404100) were two highly homologous sequences of *ANAC072 /RD26*. *PtrNAC013* gene is overexpressed in poplar 84k (*Populus alba × P. glandulosa*), and the overexpressed plants had better salt tolerance than the wild type [63]. *PdNAC013* and *PdNAC105* had significant changes in gene expression levels in leaves, roots and stems under PEG-stress treatment. *PdNAC013* shared a drought-responsive single-impulse pattern in the three tissues, which reached its peak at 72 h after PEG6000 treatment. PdNAC013 protein was localized in the nucleus in the protoplasts of NL895 poplar, which was consistent with the subcellular localization of *PtrNAC013* gene in tobacco [63]. In conclusion, *PdNAC013* and *PdNAC105* may play an important role in drought stress.

The study indicated that a few *PdNAC* genes were induced by drought, and might be implicated in poplar response to drought. However, the regulatory role of NAC genes in drought response in Populus species remains complex and elusive. For example, almost all *PtrNAC* genes had MYB binding sites (MBS) in their upstream regions, suggesting that MBS elements of a few *PtrNAC* genes might be bound by PtrMYB proteins. In wheat, TaMYBL1 protein can bind to two MBS *cis*-acting elements inserted in the promoter region of a drought-related gene *TaNAC071-A*, causing an increase in its expression level and drought tolerance in plant [64]. Thus, we will identify *cis*-acting elements and trans-acting factors, transcriptional activation domain and DNA-binding domain of poplar NAC genes associated with drought stress.

## 4. Materials and Methods

### 4.1. Genomic Data Retrieval and Poplar NAC Gene Identification

Genomic sequence (transcript and protein sequences) and GFF/GTF annotation files of *P. trichocarpa* (v3.0), *Salix purpurea* (v5.1), *Eucalyptus grandis* (v2.0) and *Vitis vinifera L* (v2.1) were downloaded from the Phytozome database (https://phytozome.jgi.doe.gov/ (accessed on 2 March 2022)). The GFF/GTF, transcript and protein files for *Arabidopsis thaliana* were retrieved from the EnsemblPlant release 53 (https://plants.ensembl.org/ (accessed on 2 March 2022)). The *A. thaliana* NAC (AthNAC) sequences were searched and acquired from the *Arabidopsis* information resource (TAIR, https://www.arabidopsis.org/ (accessed on 2 March 2022)).

To identify the *P. trichocarpa* NAC (PtrNAC) proteins, the HMM (hidden Markov model) profile of NAM domain (PF02365.15) was searched against all *P. trichocarpa* proteins by using HMMER (v3.3.2) with an E-value threshold of 1 × 10^−3^ [65]. All candidate PtrNAC proteins were aligned against AthNACs with BLASTp (v2.9.0) under an E-value cutoff of 1 × 10^−5^ [65]. These candidate proteins were further validated by searching NAC core sequences via InterProScan (https://www.ebi.ac.uk/interpro/about/interproscan/ (accessed on 8 July 2022)) and Pfam (http://pfam.xfam.org/ (accessed on 8 July 2022)) service. The length, GC-content, isoelectric point (pI), and molecular weight (MW) of all PtrNACs were calculated with Biopython (v1.79).

### 4.2. Multiple Alignment and Phylogenetic Analysis

Multiple sequence alignment (MSA) of PtrNAC protein sequences was performed via ClustalW (v2.1, http://www.clustal.org/clustal2/ (accessed on 2 March 2022)) with default parameter. The best-fitted substitution model ‘JTT+F+R7′ was inferred from the PtrNAC MSA file by ModelFinder implemented in IQ-TREE (v1.6.12) based on minimum BIC value [66]. The MSA file was used to reconstruct the maximum likelihood (ML) phylogenetic tree by using IQ-TREE with the best-fitted substitution model and 1000 bootstrap replicates [67]. The ML tree was graphically viewed with the R package ggtree (v3.2.1) [68].

### 4.3. PtNACs Gene Structures and Conserved Motifs

The PtrNACs genome database and gff3 annotation file were used to investigate the exon and intron distributions of the *PtrNAC* genes, and structure diagrams of PtrNACs were visualized by the R package ggplot2 (v3.3.6, https://github.com/tidyverse/ggplot2 (accessed on 10 March 2022)). Conserved motifs on the PtrNAC proteins were analyzed using MEME (v5.4.1, http://meme-suite.org/tools/meme (accessed on 10 March 2022)). The MEME parameters were specified as follows: the optimum motif width was between 6 and 50 in length, and the maximum number of motifs was 15.

### 4.4. Subcellular Localization Prediction and Verification

The subcellular localization of PtrNAC proteins were predicted using three web tools, including Cello (http://cello.life.nctu.edu.tw/ (accessed on 5 May 2022)), WoLFPSORT (https://wolfpsort.hgc.jp/ (accessed on 5 May 2022)) and BUSCA (http://busca.biocomp.unibo.it/ (accessed on 5 May 2022)).

In order to verify the subcellular localization of PdNAC013 (Potri.001G404100.1) in poplar NL895 (*P. deltoids* × *P. euramericana* cv. ′Nanlin895′), we did the following experiments. First, the newly expanded leaves were harvested from 8-week-old plantlets, and used for the poplar protoplast isolation with Cellulase RS (Takara, Japan, 200115-01) and Pectolyase Y23 (Takara, Japan, Y-011) according to the method of Yoo et al. [69]. Then, the recombinant plasmid *PdNAC013*::GFP (green fluorescence protein) was constructed according to homologous recombination method. D53-mCherry is a nuclear marker plasmid fused with rice protein D53 [33]. Finally, the recombinant plasmid PdNAC013::GFP and D53-mCherry was transformed into poplar protoplast through PEG-Ca^2+^ (PEG-4000) mediated transformation [69]. Its fluorescent photographs were captured on a BX53 fluorescence microscope (Olympus, Tokyo, Japan). The empty plasmid pBI221:GFP (control) was also infiltrated into the protoplasts.

### 4.5. Chromosomal Distribution and Gene Duplication Analysis

The chromosomal locations of *PtrNAC* genes were retrieved from the *P. trichocarpa* GFF annotation file, and visualized using the R package circlize (v0.4.15) [70]. Pairwise alignment of pooled protein sequences from five dicotyledonous species, including *P. trichocarpa*, *S. purpurea*, *E. grandis*, *V. vinifera* and *A. thaliana*, was carried out by BLASTp in the BLAST+ (v2.9.0) package. The resulting BLASTp outputs were used to analyze the collinearity of NAC proteins between *P. trichocarpa* and the other four species, by using Multiple Collinear Scan Toolkit (MCScanX, https://github.com/wyp1125/MCScanX (accessed on 10 March 2022)). The collinearity relationship of between *P. trichocarpa* and each of the other four species was plotted with JCVI (v0.8.12, https://github.com/tanghaibao/jcvi (accessed on 10 March 2022)). To further evaluate the characteristics of the influence of evolutionary on PtrNACs family, a simple Ka/Ks calculator was used to calculate the synonymous (Ks) and non-synonymous (Ka) of *PtrNAC* gene pairs using ParaAT (v2.0) and KaKs-Calculator (v2.0) [71]. The divergent times of duplicated poplar gene pairs were estimated with r (the rate of synonymous substitutions per site per years) = 9.1 × 10^−9^ [72,73].

### 4.6. Cis-Element Analysis

The 2 kb sequence in the front of the PtrNACs start codon (ATG) was obtained from *P. trichocarpa* genomic sequence with SeqKit (v0.8.0) [74]. The 2 kb upstream sequences were used to predict their *cis*-regulatory elements using the PlantCARE database (http://bioinformatics.psb.ugent.be/webtools/plantcare/html/ (accessed on 10 March 2022)).

### 4.7. RNA-Seq Acquisition and Expression Analysis

The RNA-seq data for the three tissues (leaf, stem and root) of *P. trichocarpa* under drought treatment were derived from NCBI SRA database (BioProject no. PRJEB19784). These RNA-seq data were pre-processed and trimmed with FastQC (v0.11.8) and Trimmomatic (v0.38). The spliced mapping of the trimmed RNA-seq reads against the *P. trichocarpa* genome (v3.0) was performed using STAR (v2.7.3a) [75]. Different expression genes (DEGs) were identified by the R package DESeq2 (v1.31.1) with a *p*-value cutoff of 0.01 [76]. The heatmap was plotted using the R package ggplot2 (v3.3.6).

### 4.8. Plant Materials and Treatments with Drought Stress

Plantlets of hybrid poplar NL895 *(P. deltoids × P. euramericana* cv. ‘Nanlin895′) were grown on the 1/2 Murashige and Skoog (MS) medium (pH = 5.8). The poplar NL895 seedlings were cultivated in a greenhouse at a room temperature (23 °C) and a relative humidity of 74%. They were cultured in 1/2 MS medium for two months and then transferred to 1/2 MS medium with 3% (*w*/*v*) PEG6000 for drought treatment. Samples were harvested at 12 h, 24 h, 36 h, 72 h and 168 h (7 days) after 3% PEG6000 treatment, respectively. These samples were immediately frozen in liquid nitrogen and stored at −80 °C for subsequent RNA isolation. The total RNA was extracted from the leaf, stem and root tissues of the poplar NL895 with plant RNA extraction kit (No. DP441, Tiangen, Beijing, China), respectively.

### 4.9. RNA Isolation and qRT-PCR

The total RNA was isolated from three vegetative organs (e.g., root, stem and leaf) of the poplar NL895 using a plant RNA extraction kit (No.DP441, Tiangen, Beijing, China). The concentration and integrity of the extracted RNA was determined using a combination of an ultraviolet-visible (UV-Vis) spectrophotometer (BioDrop, Cambridge, UK) and 1% agarose gel electrophoresis. The extracted RNA was reverse-transcribed into cDNA by a reverse transcription kit (No.AG11706, Accurate Biotechnology, Changsha, China). Then the cDNA was diluted 10 times for qRT-PCR experiment.

Prior to qRT-PCR (quantitative reverse transcription PCR), the specific primers were designed with Oligo (v7.37) using poplar CDSs (coding sequences) as reference. The primer sequences were listed in Appendix A. The ten genes were cloned using the poplar NL895 cDNA as template (Appendix A). The cloned fragments were linked to the pClone007 Blunt Vector (No.TSV-007B, TSINGKE, Beijing, China). Then the vectors were transferred to Trelief^TM^ 5α chemoreceptor cells (No.TSC-C01, TSINGKE, Beijing, China). Finally, a single colony was screened to verify the size of bands in gel-electrophoresis, and sequenced on an ABI 3730xl DNA analyzer.

Using the sequences of the ten genes as reference, the qRT-PCR primers of these *PdNAC* genes were designed by Primer Premier (v5.0) to analyze their expression patterns (Appendix A). Then, the gene expression levels were detected using ABI 7500 Fast Real-Time PCR system (Applied Biosystems, Foster City, CA, USA) and SYBR Green Premix Pro Taq HS qPCR Kit (No. AG11701, Accurate biotechnology, Changsha, China). The following conditions were used: 95 °C for 10 s, followed by 40 cycles at 95 °C for 5 s, 60 °C for 30 s, and the melting curve conditions were 60 °C for 60 s and 95 °C for 15 s. Poplar actin gene (Potri.019G006700.1) was selected as an internal reference gene. The relative expression level of *PdNAC* genes was calculated by 2^−ΔΔCt^ method [77]. Each sample was repeated three times.

## 5. Conclusions

In the research, a total of 170 *P. trichocarpa* NAC domain-containing (*PtrNAC*) genes were identified at the entire genome level. The physicochemical property calculating, chromosomal locating, phylogenetic classifying, intraspecific/interspecific collinearity inferencing, and dehydration-responsive expression profiling of these *PtrNAC* genes were performed. Analysis of *cis*-acting elements and drought-responsive expression pattern of *PtrNACs* indicated that these genes might be involved in drought stress response. In addition, ten *NAC* genes from poplar NL895 were cloned. The *PdNAC013* exhibited a tissue-specific and pulsing expression pattern under drought treatment. The subcellular localization analysis suggested that PdNAC013 protein localized in the nucleus. In summary, these findings will provide a valuable clue for further screening and characterization of *NAC* genes in relation to poplar drought response.

## Figures and Tables

**Figure 1 ijms-24-00253-f001:**
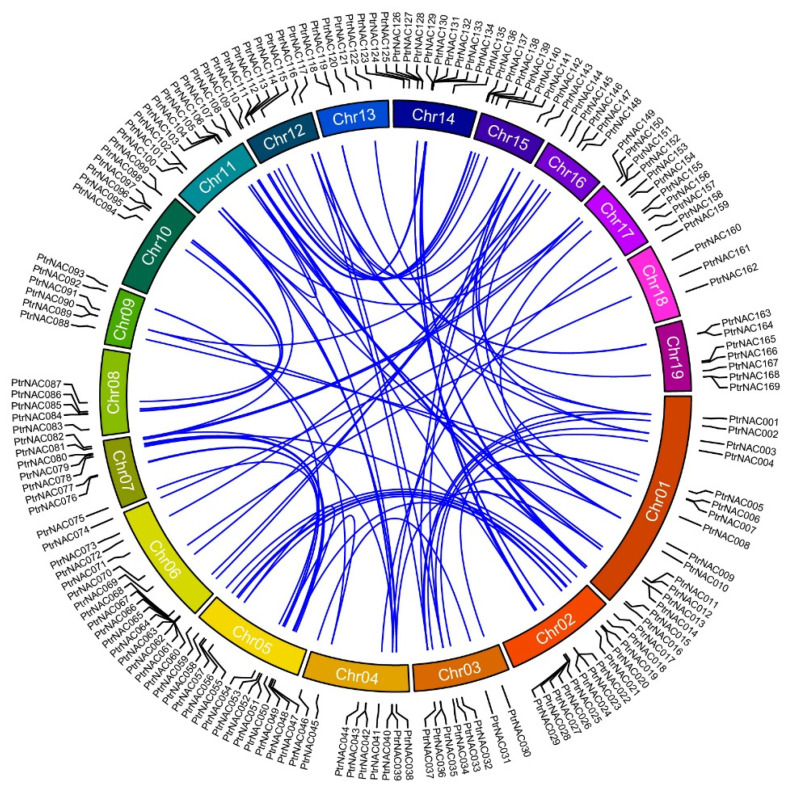
Distribution of NAC genes on the poplar genome. The blue lines inside the circle represented *PtrNAC* duplicated gene pairs, rectangles indicated chromosomes of poplar, and the gray lines denoted all poplar duplicated gene pairs.

**Figure 2 ijms-24-00253-f002:**
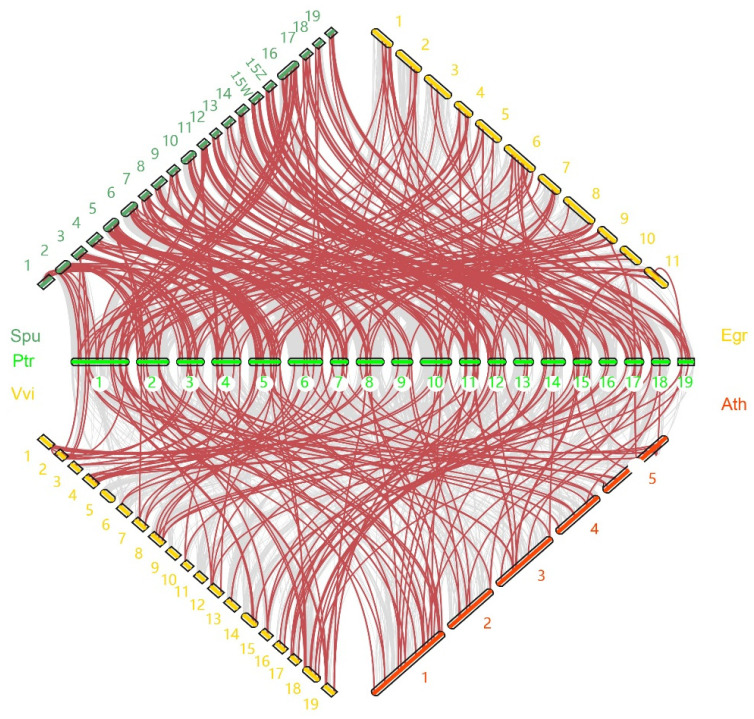
Synteny analysis of NAC genes between poplar and four plant species including *A. thaliana, E. grandis, V. vinifera and S. purpurea*. The gray lines represented collinearity of the duplicated blocks between poplar and the remaining four species, and the red lines indicated duplicated NAC gene pairs.

**Figure 3 ijms-24-00253-f003:**
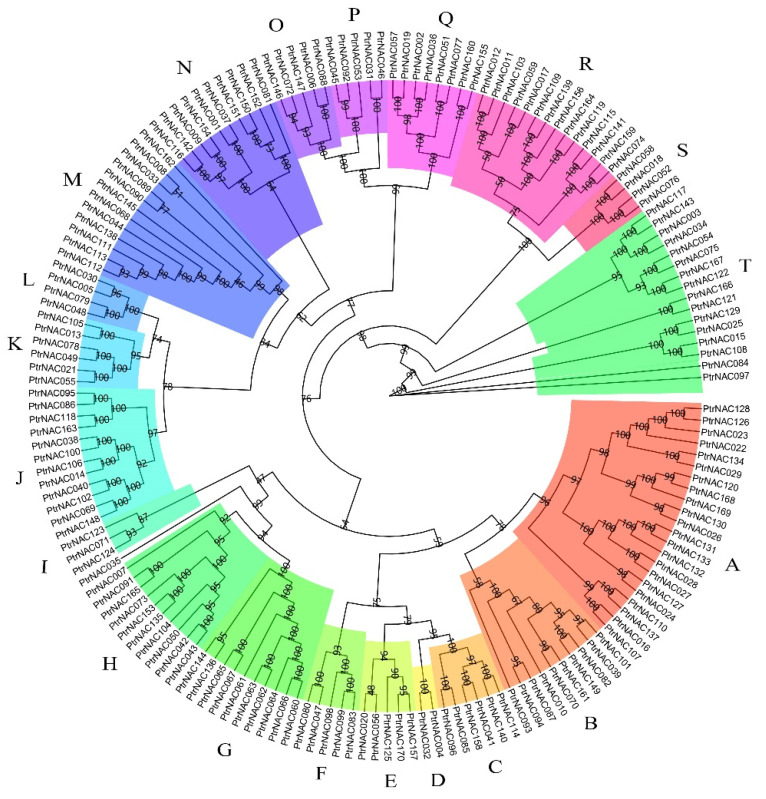
Evolutionary analysis of poplar NAC proteins. The ML (maximum likelihood) phylogenetic tree was reconstructed using IQ-TREE (v1.6.12) with ‘JTT + F+R7′ model and 1000 bootstrap replicates. The bootstrapping values were shown at the nodes in the ML tree. The twenty NAC subfamilies were marked in different colors, and their names were signed with the A–T letters.

**Figure 4 ijms-24-00253-f004:**
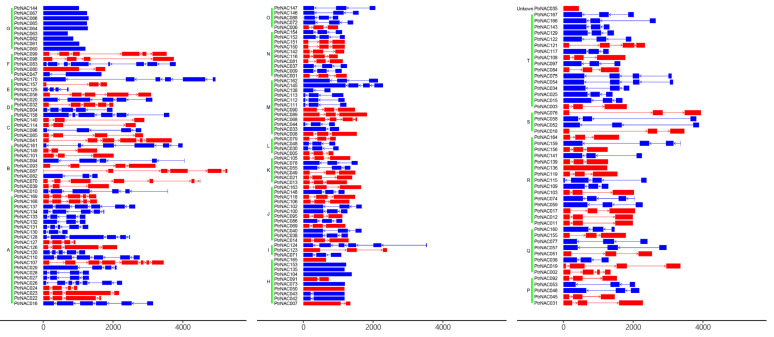
Gene structures of NAC genes in poplar. The plus-strand (+) and the minus-strand (−) of *PtrNAC* genes were marked in red and blue, respectively. Exon and intron of *PtrNACs* were indicated in a rectangle and line, respectively.

**Figure 5 ijms-24-00253-f005:**
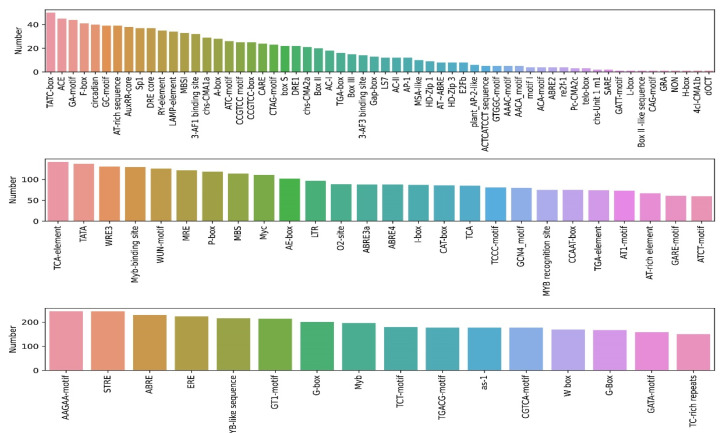
*Cis*-acting elements of *PtrNACs*. The *cis*-regulatory elements of poplar NAC genes were predicted from their 2kb upstream sequences using PlantCARE database.

**Figure 6 ijms-24-00253-f006:**
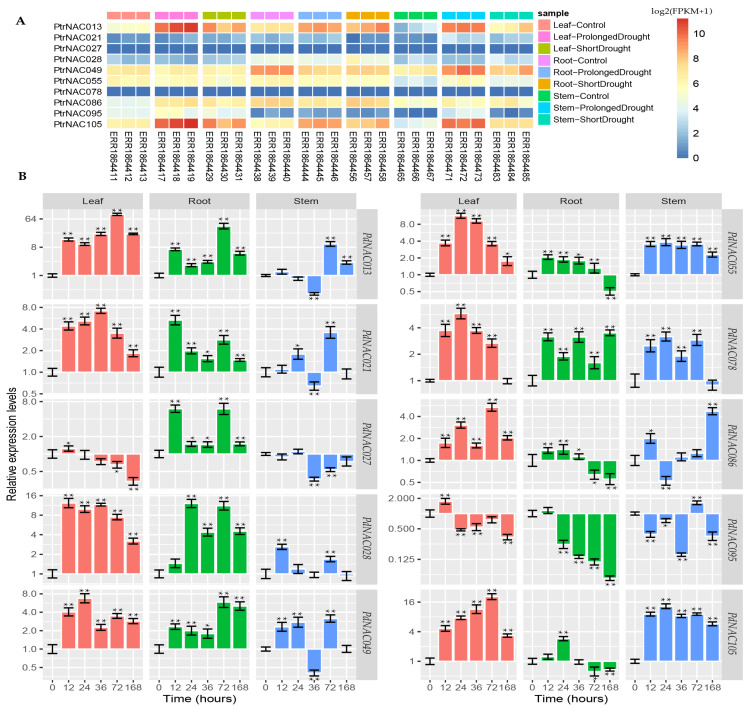
(**A**) Heatmap analysis of ten *PtrNAC* genes under drought stress. The expression value for ten *PtrNAC* genes was calculated by log2 (RPKM+1) and represented by color scale. (**B**) Tissue-specific expression patterns of ten *PdNAC* genes in poplar under drought stress. The Y-axis indicated the relative expression level. The X-axis represented hours (0, 12, 24, 36, 72 and 168) after drought treatment inseedlings of the poplar NL895. The β-actin gene (Potri.019G006700.1) was used as an internal control. The error bars were obtained from three biological replicates. The asterisks indicated that genes were significantly up-regulated or down-regulated in NL895 poplar under drought stress, according to t-test (*, *p* < 0.05; **, *p* < 0.01).

**Figure 7 ijms-24-00253-f007:**
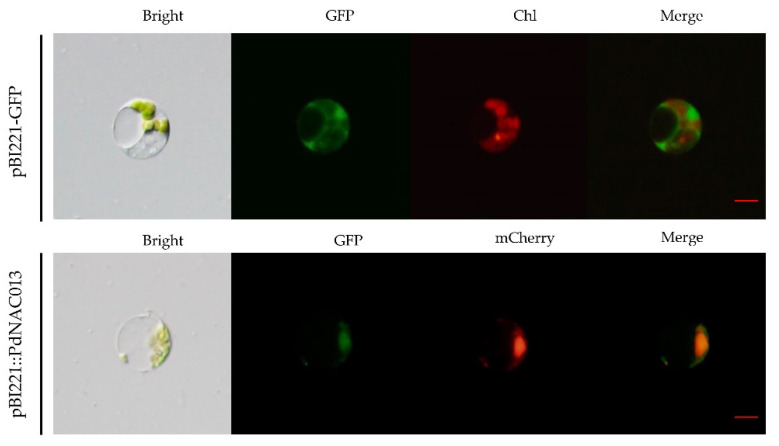
Subcellular localization of PdNAC013. The fusion protein pBI221::PdNAC13 and D53-mCherry (nuclear localization), pBI221-GFP (control) constructs were transiently expressed in NL895 poplar protoplasts and visualized with fluorescence microscope. Scale bar, 5 μm.

## Data Availability

Not applicable.

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
