# Peer review of "Genome-Wide Characterization and Evolutionary Expansion of Poplar NAC Transcription Factors and Their Tissue-Specific Expression Profiles under Drought"

_ijms, 2022, doi:10.3390/ijms24010253_

Round 1
Reviewer 1 Report
Meng et al. have done a study related to identification and characterization of NAC family members in poplar. In my opinion, the manuscript has high potential for publishing. I have some minor comments, including:
- Lines 52-53: Gene name should be in italic format. Please check the whole text.
- Introduction is well written.
- Please provide the full details of promoter element (Figure 5) in supplementary data.
- Discussion has to improve and key results should be interpreted. I suggest adding this sentence to line 258: Besides, PtrNACs showed diverse expression, indicating that modifications, such as point mutations, might occur in regulatory regions of duplicated genes, which affect the function and expression patterns (Faraji et al, 2021, Heidari et al, 2021).
References:
Faraji et al, 2021: https://doi.org/10.3390/plants10122597
Heidari et al, 2021: https://doi.org/10.3390/agronomy11081651
Author Response
Response to Reviewer 1 Comments
Thank you for the reviewers’ comments on the manuscript entitled “Genome-wide Characterization and Evolutionary Expansion of Poplar NAC Transcription Factors and Their Tissue-Specific Expression Profiles Under Drought” (ID: ijms-2090977). These comments are all valuable and helpful for revising and improving our manuscript, as well as the important guiding significance to our researches. We have studied comments carefully and have made correction which we hope meet with approval. Revised portion are marked in red and blue in the manuscript.
Point 1: Lines 52-53: Gene name should be in italic format. Please check the whole text.
Response 1: Thanks for your comment. We have revised this part italic format (lines 52-53) and checked the full text again, such as lines 282-283.
Point 2: Introduction is well written.
Response 2: Thank you very much for your recognition.
Point 3: Please provide the full details of promoter element (Figure 5) in supplementary data.
Response 3: Thank you for your suggestions. We have added the full details of promoter cis-acting elements in the Table S3.
Point 4: Discussion has to improve and key results should be interpreted. I suggest adding this sentence to line 258: Besides, PtrNACs showed diverse expression, indicating that modifications, such as point mutations, might occur in regulatory regions of duplicated genes, which affect the function and expression patterns (Faraji et al, 2021, Heidari et al, 2021).
Response 4: Thanks for your advice. We have added " Besides, PtrNACs showed diverse expression, indicating that modifications, such as point mutations, might occur in regulatory regions of duplicated genes, which affect the function and expression patterns (Faraji et al, 2021, Heidari et al, 2021)" to the discussion section (Line: 260-263, references as: [39][40]). Then we have added: ”Additionally, it is worth noting whether the young PtrNACs arisen after the divergence of Populus and Salix play a role in the adaptation of Populus species to their local environment (e.g. drought, salinity & heat). (Lines: 272-274, references as: [43,44])” Thirdly, we also improve our content, for example ”ONAC066 could bind the JBS-like (JBSL) cis-element in the promoter region of OsDREB2A gene to positively regulate drought and oxidative stress response” (Lines:284-286, references as: [53])” and ” OsNAC6, a member of ATAF family, can mediate the adaptation of plant root structure and up-regulate the expression of drought-resistant genes, which enhances the drought tolerance of rice plants. (Lines: 291-293, references as: [57-59]).
Thanks again for your help in revising and improving the manuscript.
Thank you and best regards.
Yours sincerely,
Lu Meng & Sheng Zhu

Reviewer 2 Report
Major points
Results
Figure 3: Please give the reason of using specific outgroup within the same family which may mispresent the evolutionary pattern of this gene family. Otherwise, please do not root the phylogenetic tree or use proper outgroup or midpoint it.
Figure 6 (b): Statistical significance among treatments within each facet must be calculated and plotted on the graph. Using ggplot2 variable y-axis scale function is highly recommended, for example, the high fold induction of NAC013 in the leaf shadow the expression values of the same gene in the stem.
Discussion
Should be improved
Materials and Methods
Lines 321-323: Please name the used blast algorithm and cite the reference that support the use of 1E-3 expect value to retrieve the paralogous sequences. The same applies to the following sentence.
Line 424: No need to mention the sequence of the internal reference gene as it is already mentioned in the supplementary materials.
Author Response
Response to Reviewer 2 Comments
Thank you for the reviewers’ comments on the manuscript entitled “Genome-wide Characterization and Evolutionary Expansion of Poplar NAC Transcription Factors and Their Tissue-Specific Expression Profiles Under Drought” (ID: ijms-2090977). These comments are all valuable and helpful for revising and improving our manuscript, as well as the important guiding significance to our researches. We have studied comments carefully and have made correction which we hope meet with approval. Revised portion are marked in red and blue in the manuscript.
Point 1: Figure 3: Please give the reason of using specific outgroup within the same family which may mispresent the evolutionary pattern of this gene family. Otherwise, please do not root the phylogenetic tree or use proper outgroup or midpoint it.
Response 1: Thanks for your comment. In this manuscript, we did not use an outgroup or a midpoint for phylogenetic tree analysis. In previous studies, kiwifruit NAC genes were grouped into special outgroups. Therefore, we obtained our results by referring to the literature, as shown in Figure 3.
References as: https://doi.org/10.1186/s12870-020-02798-2.
Point 2: Figure 6 (b): Statistical significance among treatments within each facet must be calculated and plotted on the graph. Using ggplot2 variable y-axis scale function is highly recommended, for example, the high fold induction of NAC013 in the leaf shadow the expression values of the same gene in the stem.
Response 2: We feel great thanks for your professional review work on our article. We have calculated the statistical significance according to the reviewer's comments and redrew Figure 6 (B). This is shown in the figure below:
(B) Tissue-specific expression patterns of ten PdNAC genes in poplar under drought stress. The Y-axis indicated the relative expression level. The X-axis represented hours (0, 12, 24, 36, 72 and 168) after drought treatment in seedlings of the poplar NL895. The β-actin gene (Potri.019G006700.1) was used as an internal control. The error bars were obtained from three biological replicates. The asterisks indicate that genes were significantly up-regulated or down-regulated in NL895 poplar under drought stress, according to t-test (*, p < 0.05; **, p < 0.01).
Point 3: Discussion: Should be improved
Response 3: As the reviewer can see, we put a lot of thought into the discussion. We agree with this suggestion and have added content of discussion based on helpful comments from reviewer. First of all, we added the content:” Besides, PtrNACs showed diverse expression, indicating that modifications, such as point mutations, might occur in regulatory regions of duplicated genes, which affected the function and expression patterns.” (Line: 260-263, references as: [39][40]). Secondly, we added: ”Additionally, it is worth noting whether the young PtrNACs arisen after the divergence of Populus and Salix play a role in the adaptation of Populus species to their local environment (e.g. drought, salinity & heat). (Lines: 272-274, references as: [43,44])” Then, we also added: ”ONAC066 could bind the JBS-like (JBSL) cis-element in the promoter region of OsDREB2A gene to positively regulate drought and oxidative stress response. (Lines:284-286, references as: [53]).” and “OsNAC6, a member of ATAF family, can mediate the adaptation of plant root structure and up-regulate the expression of drought-resistant genes, which enhances the drought tolerance of rice plants. (Lines: 291-293, references as: [57-59])”
Point 4: Lines 321-323: Please name the used blast algorithm and cite the reference that support the use of 1E-3 expect value to retrieve the paralogous sequences. The same applies to the following sentence.
Response 4: Thanks for your comment. We have cited the relevant literature at the appropriate location. In addition, HMMER (http://hmmer.org/) is a tool for protein domain searching, but not BLAST (Lines 335-337, references as: [65]).
Point 5: Line 424: No need to mention the sequence of the internal reference gene as it is already mentioned in the supplementary materials.
Response 5: Thank you for your suggestions. We have deleted the primer sequences of internal reference gene (Line 424).
Thanks again for your help in revising and improving the manuscript.
Thank you and best regards.
Yours sincerely,
Lu Meng & Sheng Zhu
